# The leak channel NALCN controls tonic firing and glycolytic sensitivity of substantia nigra pars reticulata neurons

Andrew Lutas[1], Carolina Lahmann[1], Magali Soumillon[2], Gary Yellen[1]*

[1]Department of Neurobiology, Harvard Medical School, Boston, United States;
[2]Broad Institute, Cambridge, United States

**Abstract** Certain neuron types fire spontaneously at high rates, an ability that is crucial for their function in brain circuits. The spontaneously active GABAergic neurons of the substantia nigra pars reticulata (SNr), a major output of the basal ganglia, provide tonic inhibition of downstream brain areas. A depolarizing 'leak' current supports this firing pattern, but its molecular basis remains poorly understood. To understand how SNr neurons maintain tonic activity, we used single-cell RNA sequencing to determine the transcriptome of individual mouse SNr neurons. We discovered that SNr neurons express the sodium leak channel, NALCN, and that SNr neurons lacking NALCN have impaired spontaneous firing. In addition, NALCN is involved in the modulation of excitability by changes in glycolysis and by activation of muscarinic acetylcholine receptors. Our findings suggest that disruption of NALCN could impair the basal ganglia circuit, which may underlie the severe motor deficits in humans carrying mutations in NALCN.

## Introduction

Some neurons are capable of continuously firing action potentials in the complete absence of synaptic input (*Häusser et al., 2004*). This spontaneous activity is established by the intrinsic membrane properties that set the neuronal membrane potential and support repeated firing of action potentials (*Bean, 2007*). Electrophysiological studies of spontaneously active neurons have frequently reported the presence of a tonic current that maintains these neurons at a more depolarized membrane potential and likely allows them to continuously fire action potentials (*Atherton and Bevan, 2005*; *Raman et al., 2000*; *Jackson et al., 2004*; *Taddese and Bean, 2002*; *Khaliq and Bean, 2010*). This current, often referred to as a 'leak' or 'background' current, has characteristics of a nonselective cation current – it is sodium-dependent and has a reversal potential close to 0 mV. However, the molecular identity of the channel(s) that carry this leak current in many spontaneously firing neurons remains an open question.

In the substantia nigra pars reticulata (SNr), GABAergic neurons fire spontaneously at relatively high firing rates, typically 30 spikes per second (*Sanderson et al., 1986*; *Gulley et al., 1999*; *Maurice et al., 2003*; *Deransart et al., 2003*). This spontaneous firing is necessary for the tonic inhibition of downstream targets of the SNr and thus critical for the function of the basal ganglia (*Chevalier and Deniau, 1990*; *Hikosaka, 2007*). SNr neurons have a leak current (a sodium-dependent tonic current) that maintains their membrane potential sufficiently depolarized to allow for spontaneous triggering of action potentials in the absence of any synaptic input (*Atherton and Bevan, 2005*; *Zhou and Lee, 2011*). It has been suggested that a member of the Transient Receptor Potential (TRP) channel family, TRPC3, is the basis for the tonic depolarizing current in SNr neurons (*Zhou et al., 2008*). However, we previously found that the spontaneous firing of SNr neurons was unaffected by genetic deletion of TRPC3, arguing against this possibility (*Lutas et al., 2014*). We

**\*For correspondence:**
gary_yellen@hms.harvard.edu

**Competing interests:** The authors declare that no competing interests exist.

**eLife digest** Some neurons in the brain produce electrical signals (or "fire") spontaneously, without receiving any other signals from the senses or from other neurons. This spontaneous activity has a number of important roles. For example, in a part of the brain known as the substantia nigra pars reticulata (SNr), spontaneously active neurons frequently produce electrical signals that reduce electrical activity in other brain areas.

A current of positively charged ions constantly flows into the spontaneously active SNr neurons and enables them to fire constantly. Ions enter neurons through proteins called ion channels that are embedded in the surface of the neuron. Like all proteins, ion channels are made by "transcribing" genes to form molecules of RNA that are then "translated" to produce the basic sequence of the protein.

Lutas et al. have now used single-cell RNA sequencing to study SNr neurons from mice and investigate which ion channel the positive ion current flows through. The RNA sequences revealed that the neurons have the gene for an ion channel known as NALCN. Recordings of the firing rate of neurons in slices of mouse brain showed that SNr neurons without this channel did not fire as often as SNr neurons with the channel. In addition, neurotransmitters (chemicals that alter the ability of neurons to fire) and changes in cell metabolism had less of an effect on the firing rate of SNr neurons that lacked the NALCN channel than they do on normal neurons.

These findings may help explain why people with mutations in the NALCN gene have movement disorders, as the substantia nigra pars reticulata plays an important role in orchestrating complex movements. Future work is now needed to understand how a change in NALCN activity affects the other brain areas that SNr neurons connect to.

also previously reported that the tonic depolarizing current in SNr neurons is decreased in response to metabolic inhibition, resulting in slowed firing (*Lutas et al., 2014*). Here we report our efforts to identify the nonselective cation channel (NSCC) that supports the spontaneous firing of SNr neurons and determine the molecular identity of the glycolysis-sensitive NSCC.

We used single-cell RNA sequencing to determine the gene expression profile of SNr neurons and, specifically, the relative expression levels of NSCCs. We found that the sodium leak channel, NALCN (*Lu et al., 2007*), was the highest expressed NSCC in SNr neurons. Electrophysiological experiments using pharmacological inhibitors supported a critical function of NALCN in maintaining firing, and conditional knockout of NALCN in SNr neurons impaired firing, confirming its importance. In addition, we found that the loss of NALCN significantly impaired modulation of SNr firing by glycolytic inhibition and by muscarinic metabotropic receptor activation, indicating that NALCN was involved in the regulation of SNr neuron excitability. Recent studies have reported that NALCN is expressed by other neurons in the brain (*Lu et al., 2007*; *Flourakis et al., 2015*; *Goldberg et al., 2012*), indicating that our findings in the SNr may be more broadly applicable. Furthermore, our results identify key cellular responses that are impaired in the absence of NALCN, and may have therapeutic value as human mutations in NALCN lead to severe motor and cognitive deficits (*Chong et al., 2015*; *Aoyagi et al., 2015*; *Al-Sayed et al., 2013*; *Köroğlu et al., 2013*; *Fukai et al., 2016*).

## Results

### A NSCC is responsible for the spontaneous firing of SNr neurons

A tetrodotoxin-insensitive, tonic depolarizing current is necessary for spontaneous firing of SNr neurons, and this current has been attributed to constitutive activity of a NSCC (*Atherton and Bevan, 2005*; *Zhou et al., 2008*). We previously found that inhibition of glycolysis in SNr neurons leads to a decrease in a tonic, depolarizing current and slows SNr firing (*Lutas et al., 2014*). We thus sought to identify the ion channel that carries this glycolysis-dependent tonic depolarizing current in SNr neurons. We initially used pharmacological tools that nonspecifically block multiple types of NSCCs. Flufenamic acid (FFA; 100 µM), a commonly used NSCC blocker, rapidly decreased SNr firing

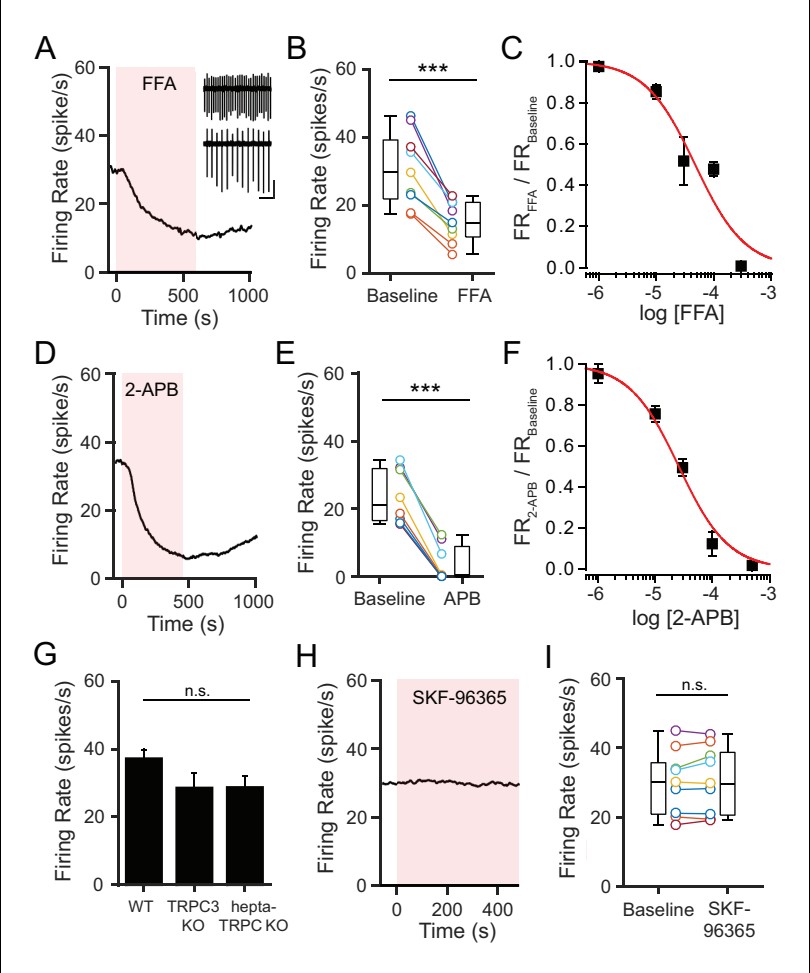

**Figure 1.** NSCCs other than TRPC channels maintain the spontaneous firing of SNr neurons. (**A,D**) Representative time course of SNr neuron firing rate with application of NSCC blocker flufenamic acid (FFA; 100 µM) (**A**) and 2-aminoethoxydiphenylborane (2-APB; 100 µM) (**D**). Inset in panel **a** shows 1 s recordings before and after FFA. Scale bar: 100 pA, 200 ms. (**B,E**) Scatter plot of mean firing rate of SNr neurons before and after FFA application (30.7 ± 3.7 versus 15.2 ± 2.0 spikes/s; paired t-test; p = 0.00013; n = 9) (**B**) and before and after 2-APB (23.6 ± 2.8 versus 3.9 ± 1.9 spikes/s; paired t-test; p<0.0001; n = 8) (**E**). Box plots indicate the population median, interquartile range, and maximum and minimum values. (**C,F**) Dose response of FFA (**C**) and 2-APB (**F**) on firing rate. Data are fit with a Hill function (red line). (**G**) Mean basal firing rate of SNr neurons from wild-type mice (WT; 37.3 ± 2.7; n = 20), mice lacking TRPC3 (TRPC3 KO; 28.6 ± 4.3; n = 10) and mice lacking all seven TRPC channels (hepta-TRPC KO; 28.7 ± 3.4; n = 7). Error bars indicate s.e.m. [n.s.]p>0.05; one-way ANOVA. (**H**) Representative time course of SNr firing rate with application of TRPC channel blocker SKF-96365 (100 µM). (**I**) Mean firing rate of SNr neurons before and after SKF-96365 application (30.0 ± 3.1 versus 30.7 ± 3.2 spikes/s; paired t-test; p = 0.19; n = 9), and box plot summary of the population statistics before and after SKF-96365. For panels B, E, I: [n.s.]p>0.05; ***p<0.001.

(*Figure 1A,B*). The effect of FFA on SNr firing was dose dependent, and at higher concentrations, FFA could even silence SNr firing ($K_d$ = 4.7 × $10^{-5}$10-5 M; *Figure 1C*). Similarly, 2-aminoethoxydiphenylborane (2-APB; 100 µM), which also blocks multiple NSCCs including many TRP channels (*Lievremont et al., 2005*; *Chen et al., 2012*), decreased SNr firing (*Figure 1D,E*). The effect of 2-APB on SNr firing was also dose dependent and capable of silencing firing at sufficiently high concentrations ($K_d$ = 2.6 × $10^{-5}$ M; *Figure 1F*). These results are consistent with earlier studies and support an important role of NSCCs in maintaining the spontaneous firing of SNr neurons (*Zhou and Lee, 2011*).

Based on previous work, TRPC3 has been proposed as the molecular basis of the tonic depolarizing current in SNr neurons. Zhou and coworkers reported that SNr neurons express TRPC3 but no

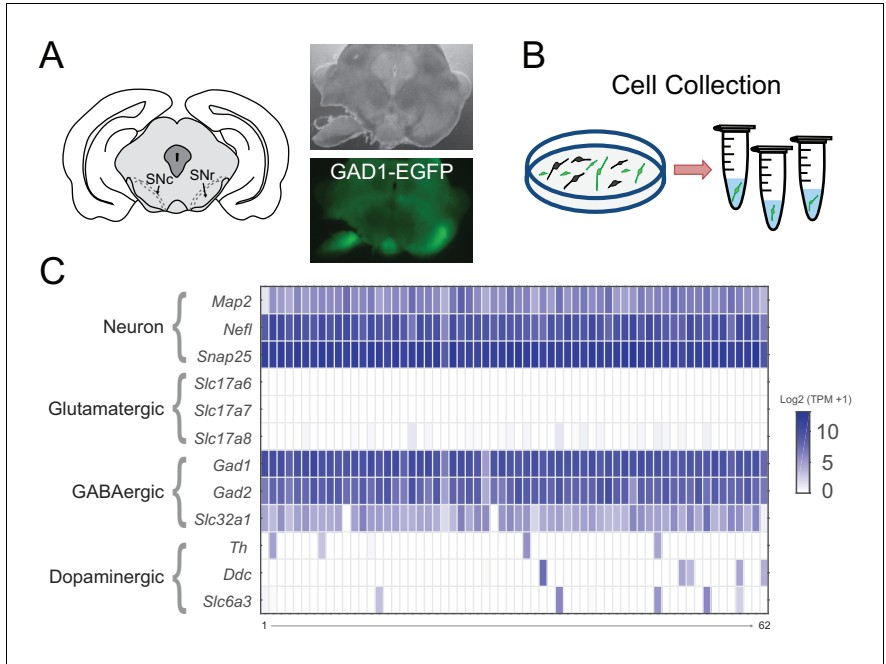

**Figure 2.** Transcriptome sequencing of individual SNr GABAergic neurons. (**A**) (*left*) Drawing depicting a coronal section of the mouse brain containing the substantia nigra pars compacta (SNc) and SNr. Bright field (*right top*) and green fluorescent (*right bottom*) images of a coronal brain section from a GAD1-EGFP mouse with the substantia nigra region of one hemisphere partially microdissected. (**B**) Scheme for the plating and collecting of individual GFP-positive neurons. (**C**) Gene expression results for the 62 samples that were of high quality. The rows indicate gene expression results for neuron specific marker genes (*Map2, Nefl, Snap25*) and marker genes for neuronal subtypes: glutamatergic (*Slc17a6, Slc17a7, Slc17a8*), GABAergic (*Gad1, Gad2, Slc32a1*), and dopaminergic (*Th, Ddc, Slc6a3*). Expression is indicated by the log2 transform of the number of transcripts per million (TPM) plus 1.

other TRP channels, and that the constitutive activity of TRPC3 is necessary for the spontaneous firing of SNr neurons (*Zhou et al., 2008*). Surprisingly, however, we previously found that SNr neurons from mice lacking TRPC3 continue to fire spontaneously at high rates that were not significantly different from wild-type (WT) SNr neurons (data from *Lutas et al., 2014*; *Figure 1G*). Furthermore, the firing rate of SNr neurons lacking all seven TRPC members (TRPC1-7) was not significantly attenuated either, arguing against compensation by other TRPC channels in the TRPC3 knockout animals (data from *Lutas et al., 2014*; *Figure 1G*). Consistent with this, an inhibitor of TRPC channels, SKF-96365 (100 µM), did not decrease the firing rate of SNr neurons (*Figure 1H,I*). Taken together, these results indicate that TRPC channels are not required to maintain the spontaneous firing of SNr neurons and are unlikely to provide the basis for the tonic depolarizing current.

## RNA sequencing of individual SNr neurons

To clarify which NSSC are expressed by SNr neurons, we chose an unbiased approach that took advantage of advances in RNA sequencing (RNA-seq) technology to obtain whole transcriptome gene expression data from individual SNr neurons. This method required the isolation of GABAergic neurons that reside within the SNr from other cell types, especially the neighboring dopaminergic neurons. While currently there is no selective marker for SNr neurons, we could effectively isolate SNr neurons by a combination of anatomical localization and fluorescent marking of GABAergic neurons. We made acute coronal brain slices containing the SNr from brains of GAD1-EGFP transgenic mice, in which all GABAergic neurons are fluorescently labeled (*Figure 2A*). Then, we dissected out the SNr regions from brain slices, dissociated this tissue to isolate individual cells, and manually collected GFP-positive neurons with a micromanipulator while observing the cells in a fluorescent microscope (*Figure 2B*). We collected 87 GFP-positive SNr neurons, thus isolating GABAergic SNr

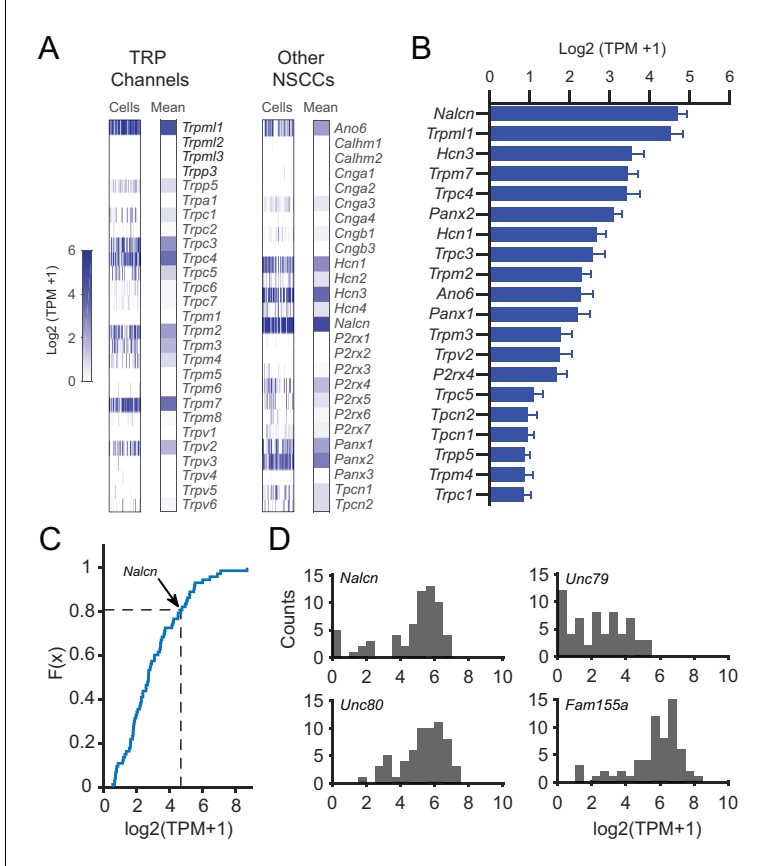

**Figure 3.** Gene expression results for NSCCs in SNr neurons. (**A**) Relative gene expression levels of TRP channels (*left*) and other NSCCs (*right*) in individual SNr neurons and the population average. (**B**) Bar graph of the mean log2 (TPM +1) for the top 20 NSCCs in descending order. Error bars indicate s.e.m. (**C**) Cumulative distribution plot of the mean expression level of all ion channels that were consistently expressed by SNr neurons (73 out of 232 total ion channel genes). The average level of expression of *Nalcn* is indicate by arrow and dashed line. (**D**) Histograms of expression of *Nalcn, Unc79, Unc80,* and *Fam155a* in SNr neurons.

The following figure supplement is available for figure 3:

**Figure supplement 1.** SNr neuron gene expression results for all ion channels.

neurons from neighboring dopaminergic neurons of the substantia nigra pars compacta and also separating them from dopamine neurons located within the SNr. To improve our ability to detect ion channel genes, which have comparatively low expression levels, we employed the SMART-seq2 method for preparing samples for RNA sequencing, which allows for reads to be made across the full sequence of gene transcripts (*Picelli et al., 2014*).

To ensure high-quality data, we focused our analysis on samples that had greater than 5000 genes detected (7955 ± 170; n = 62). We first confirmed the selectivity of our SNr neuron RNA sequencing by examining cell-type marker genes (*Figure 2C*). GFP-positive cells expressed general neuronal markers *Map2, Nefl,* and *Snap25*. As expected, they also expressed GABAergic neuron markers *Gad1, Gad2* (glutamic acid decarboxylases required for GABA synthesis) and *Slc32a1* (the vesicular GABA transporter) (*Figure 2C*). In contrast, these cells had little or no expression of dopaminergic neuron markers *Th* (tyrosine hydroxylase required for the synthesis of a dopamine precursor), *Ddc* (a decarboxylase required for dopamine synthesis), and *Slc6a3* (the dopamine transporter) (*Figure 2C*). There was also little to no expression of glutamatergic neuronal markers *Slc17a6, Slc17a7,* and *Slc17a8* (vesicular glutamate transporters) (*Figure 2C*). We were therefore able to

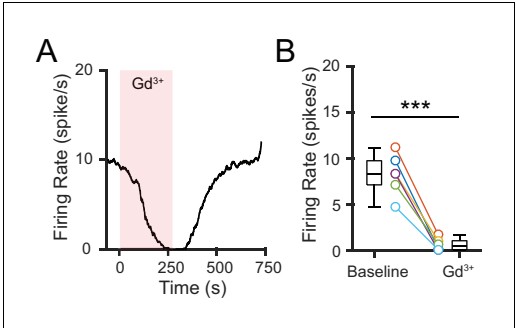

**Figure 4.** Pharmacological blockade of NALCN impairs SNr firing. (**A**) Representative time course of SNr neuron firing rate with application of gadolinium (Gd$^{3+}$; 100 µM). (**B**) Comparison of firing rates before and after Gd$^{3+}$ application (8.2 ± 0.9 versus 0.6 ± 0.3 spikes/s; paired t-test; $p$=0.00018; $n$ = 6). ***p<0.001.

obtain whole transcriptome gene expression results from GABAergic SNr neurons, which guided our search for the NSCC that maintains the spontaneous firing of SNr neurons.

## SNr neurons express several NSCCs

To generate a candidate list of NSCCs in SNr neurons, we examined the expression of known NSCCs. In contrast to a previous study, we found that SNr neurons express several members of the TRP channel family in addition to TRPC3 (*Figure 3A*). Interestingly, we found that NSCCs other than TRP channels were also expressed by SNr neurons. In particular, NALCN stood out as a promising candidate for the tonic depolarizing current, as it had the highest expression of all NSCCs (*Figure 3B*) and its expression was in the top 20% of all ion channels (*Figure 3C* and *Figure 3—figure supplement 1*). In addition, we found that SNr neurons express *Unc80, Unc79,* and *Fam155a*, three genes whose protein products are known to interact with NALCN (*Xie et al., 2013*; *Ren, 2011*) (*Figure 3D*). We therefore hypothesized that the spontaneous firing rate of SNr neurons requires functional NALCN.

## NALCN is a component of the machinery that sustains the spontaneous firing of SNr neurons

We first investigated whether NALCN is functionally active in SNr neurons using the trivalent cation gadolinium (Gd$^{3+}$), which has been shown to block NALCN (*Lu et al., 2007*). To prevent precipitation of Gd$^{3+}$ salts, we used a modified ACSF that was buffered by HEPES instead of sodium bicarbonate and sodium phosphate; the basal firing rate of SNr neurons in this modified ACSF is ~ 60% lower than that in the standard ACSF. Application of Gd$^{3+}$ (100 µM) rapidly silenced the firing of SNr neurons, suggesting that NALCN may be functionally active in these neurons (*Figure 4A,B*).

Given that Gd$^{3+}$ is not a specific blocker and may affect other channels, blockade of channels other than NALCN may be contributing to the effects of Gd$^{3+}$ on firing rate. To overcome the caveats inherent in the use of nonspecific pharmacological blockers, we used a genetic approach to directly test whether NALCN is necessary for the spontaneous activity of SNr neurons. We used transgenic mice in which an exon of *Nalcn* is flanked by loxP sites (*Nalcn[fl/fl]*) (*Flourakis et al., 2015*) to conditionally eliminate NALCN in SNr neurons. The cre-dependent knockout was induced by unilateral injection of an adeno-associated virus (AAV) vector expressing fluorescently tagged cre recombinase driven by a neuron-specific promoter (hSyn-EGFP-cre) (*Figure 5A*). After expression for 1–2 weeks, acute brain slices were prepared and GFP-positive (NALCN KO) cells were targeted for cell-attached patch clamp recordings (*Figure 5B*). GFP-negative (control) cells from the injected and uninjected hemispheres were used as controls. Consistent with an important role of NALCN in maintaining spontaneous firing of SNr neurons, the spontaneous firing rate of NALCN KO SNr neurons was significantly lower than control SNr neurons (*Figure 5C,D*). In contrast, the firing rate of virally transduced SNr neurons from WT mice was unaffected, confirming that the slower firing rate of *Nalcn[fl/fl]* SNr neurons transduced with EGFP-cre was not a consequence of the viral transduction (*Figure 5D*).

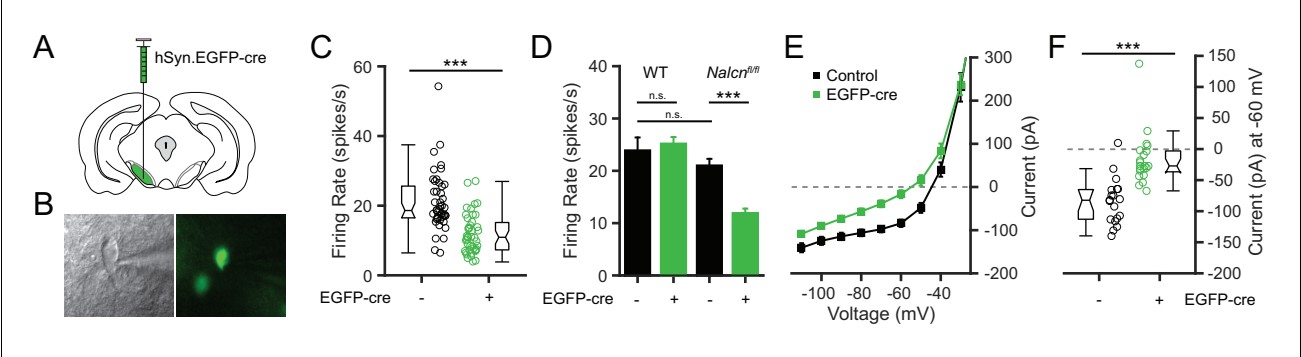

**Figure 5.** Conditional knockout of NALCN in SNr neurons slows spontaneous firing. (**A**) Schematic depicting unilateral injections of the adeno-associated virus containing the hSyn-EGFP-cre construct. (**B**) Bright field (*left*) and fluorescent (*right*) images of an example EGFP-cre positive SNr neuron from which recordings were obtained. (**C**) Firing rates of EGFP-cre negative ($21.0 \pm 1.3$ spikes/s; $n = 45$) versus positive ($11.9 \pm 0.9$ spikes/s; $n = 43$) neurons from *Nalcn*[fl/fl] transgenic mice (unpaired t-test; ***$p<0.0001$). Box plot notches indicate the 95% confidence intervals. (**D**) Mean firing rate of EGFP-cre negative ($23.9 \pm 2.5$ spikes/s; $n = 27$) and positive ($25.2 \pm 1.3$ spikes/s; $n = 39$) neurons from WT mice (unpaired t-test; $P = 0.62$), and EGFP-cre negative and positive neurons from *Nalcn*[fl/fl] mice. [n.s.]$p>0.05$; ***$p<0.001$; one-way ANOVA with Bonferroni *post hoc* correction. (**E**) Steady-state I-V recordings from EGFP-cre negative (black symbols; $n = 19$) and positive (green; $n = 21$) neurons. (**F**) Steady-state current (pA) at -60 mV from EGFP-cre negative ($n = 19$) and positive ($n = 21$) neurons (unpaired t-test; ***$p<0.0001$).

To confirm that the NALCN-mediated inward leak current was disrupted in the NALCN KO neurons, we performed whole-cell voltage-clamp recordings to measure steady-stead inward currents. As predicted, NALCN KO neurons showed reduced inward currents compared to control SNr neurons (*Figure 5E*). At $-60$ mV, which is about the typical membrane potential during interspike intervals and the steady-state resting potential of SNr neurons when action potentials are blocked (*Lutas et al., 2014*), NALCN KO neurons had significantly less inward current compared to control SNr neurons (*Figure 5F*). These data confirm that a decreased steady-state inward current mediates the slowing in firing rate observed for NALCN KO neurons.

## NALCN contributes to the glycolysis-sensitivity of SNr firing

The spontaneous activity of SNr neurons is modulated by cellular metabolism (*Lutas et al., 2014*; *Yamada, 2001*; *Ma et al., 2007*). We previously reported that inhibiting glycolysis while in the continued presence of the mitochondrial fuel beta-hydroxybutyrate (βHB) slows the firing rate of SNr neurons by a mechanism that involves decreasing a tonically active nonselective cation current (*Lutas et al., 2014*). We therefore tested the effect of glycolytic inhibition on SNr neurons lacking NALCN, to learn whether NALCN underlies the glycolysis-sensitive nonselective cation current. We measured spontaneous firing rates of control and NALCN KO SNr neurons, and tested the effect of inhibition of glycolysis in the presence of the mitochondrial fuel beta-hydroxybutyrate (βHB).

Control SNr neurons showed a robust decrease in firing rate with inhibition of glycolysis using iodoacetic acid (IAA; 1 mM), similar to our previous results (*Lutas et al., 2014*) (*Figure 6A,B*). In contrast, NALCN KO SNr neurons showed only a modest change in firing rate after inhibition of glycolysis (*Figure 6A,B*). We wondered whether the smaller effect of IAA on NALCN KO neurons might simply be due to the lower initial firing rate, because of a non-linearity in the relationship between firing rate and the ionic current. We thus used elevated extracellular [K+] (from 2.5 to 4 mM) to increase the basal firing rate of NALCN KO cells to approximately the firing rate of control cells. In increased extracellular [K+] bath solution, NALCN KO neurons still fired significantly slower than control neurons (*Figure 6—figure supplement 1A*), and in both potassium concentrations, NALCN KO neurons fired about 60% slower than control neurons (*Figure 6—figure supplement 1B*).

NALCN KO neurons recorded in either of the two potassium concentrations had significantly smaller reductions in firing rate in response to glycolytic inhibition compared to control neurons ($\sim$20% versus $\sim$60%), indicating that the difference was not simply a result of the lower starting firing rate of NALCN KO SNr neurons (*Figure 6B,C*). Together, these data suggest that NALCN is a major component of the glycolysis-sensitive nonselective cation current in SNr neurons.

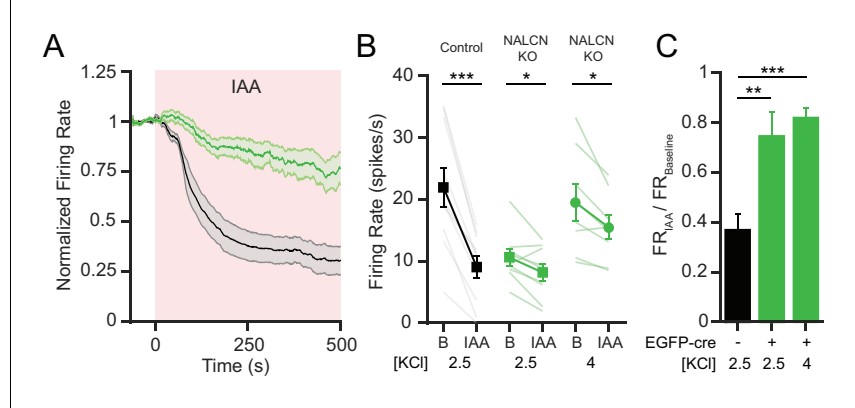

**Figure 6.** NALCN is required for the decrease in SNr firing rate after inhibition of glycolysis. (**A**) Time course of the average firing rate of control (GFP-negative; black line) and NALCN KO neurons (GFP-positive; green line) with application of glycolytic inhibitor, IAA (1 mM). Shaded error region indicates s.e.m. (**B**) Mean firing rate of SNr neurons during baseline and after application of IAA. Black square symbols: GFP-negative control experiments in 2.5 mM KCl bath solution (21.9 ± 3.2 versus 9.1 ± 1.8 spikes/s; paired t-test; p<0.0001; *n* = 10). Green square symbols: NALCN KO experiments in 2.5 mM KCl bath solution (10.6 ± 1.3 versus 8.1 ± 1.4 spikes/s; paired t-test; p=0.03; *n* = 9). Green circle symbols: NALCN KO experiments in 4 mM KCl bath solution (19.5 ± 3.0 versus 15.5 ± 2.0 spikes/s; paired t-test; p=0.02; *n* = 8). *p<0.05; ***p<0.001. (**C**) Baseline normalized firing rate showing the fold change after application of IAA in control (2.5 mM KCl) and both NALCN KO conditions (2.5 and 4 mM KCl). **p<0.01; ***p<0.001; one-way ANOVA with Bonferroni *post hoc* correction.
The following figure supplement is available for figure 6:

**Figure supplement 1.** NALCN KO SNr neurons have lower firing rates in both low and high potassium conditions.

## The response of SNr neurons lacking NALCN to muscarinic receptor activation is blunted

Spontaneous firing of SNr neurons can also be modulated by other signaling processes, particularly activation of metabotropic receptors (*Zhou and Lee, 2011*). It has been postulated that metabotropic receptor-mediated changes in SNr excitability required the activity of TRPC3 channels (*Zhou and Lee, 2011*; *Zhou et al., 2009*). Interestingly, metabotropic receptor activation can also modulate NALCN activity (*Swayne et al., 2009*; *Kim et al., 2012*). We tested whether TRPC3 and NALCN underlie the changes in SNr firing in response to metabotropic receptor activation. We focused on muscarinic receptor activation as this receptor has been shown to activate NALCN channels in pancreatic beta-cells (*Swayne et al., 2009*).

Activation of muscarinic acetylcholine receptors (mAChR) using oxotremorine-M (Oxo-M; 10 µM) rapidly increased SNr firing rate by over two-fold (*Figure 7A*). In contrast, the effect of Oxo-M on the firing rate of SNr neurons lacking NALCN was less prominent (*Figure 7A*). In control neurons, the maximum firing rate in Oxo-M was significantly greater than the basal firing rate (*Figure 7B*). While NALCN KO SNr neurons also increased their firing rate in response to Oxo-M (*Figure 7B*), the normalized peak response to Oxo-M was substantially smaller than in control SNr neurons (*Figure 7C*). For both the control and NALCN KO neurons, no significant correlation existed between the response to Oxo-M and the basal firing rate, indicating that the difference between control and NALCN KO responses to Oxo-M was not explained by the different starting firing rates (*Figure 7D*).

Contrary to the results from NALCN KO neurons, SNr neurons lacking TRPC3 or all seven TRPC channels still exhibited large increases in firing rate after application of Oxo-M, similar to WT SNr neurons (*Figure 7E*). The peak fold increase of WT, TRPC KO, and KO of all seven TRPC was not significantly different (*Figure 7F*). These results indicate that NALCN, but not TRPC3, is important for the regulation of SNr neuron excitability by mAChR activation.

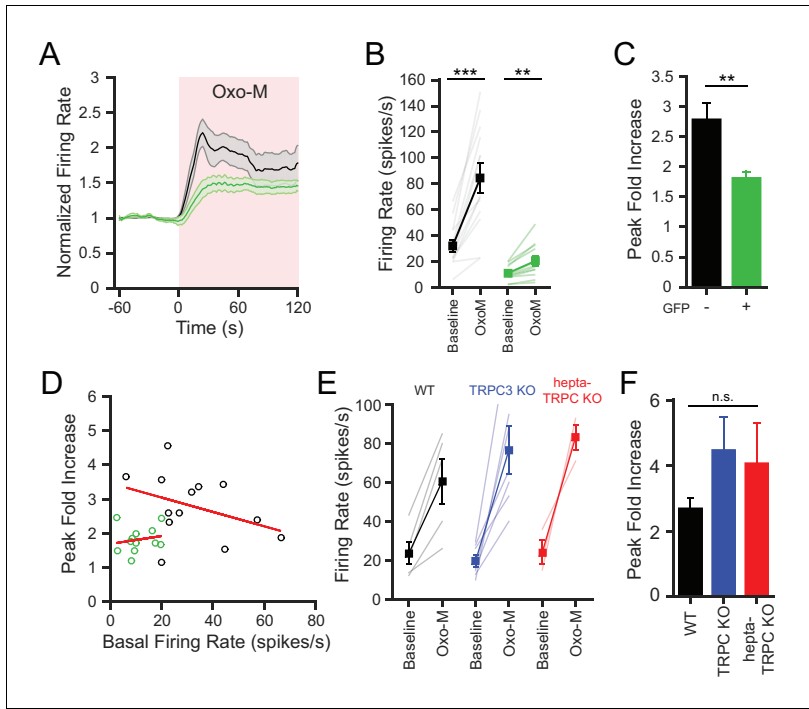

**Figure 7.** Conditional knockout of NALCN in SNr neurons blunts the increased excitability with activation of metabotropic acetylcholine receptors. (**A**) Time course of the baseline normalized mean firing rate of control and NALCN KO SNr neurons with application of Oxo-M (10 µM). Shaded error region indicates s.e.m. (**B**) Mean firing rate during the baseline period versus the maximum firing rate during Oxo-M application of control (84.5 ± 11.4 versus 32.3 ± 4.7 spikes/s; paired t-test; p<0.0001; $n$ = 13) and NALCN KO SNr neurons (20.6 ± 3.8 versus 11.1 ± 1.7 spikes/s; paired t-test; $p$=0.002; $n$ = 12). **p<0.01; ***p<0.001. (**C**) Peak firing rate during Oxo-M normalized to baseline firing rate of control versus NALCN KO neurons (2.8 ± 0.3 versus 1.8 ± 0.1; paired t-test; $p$=0.008). **p<0.01. (**D**) Scatter plot of the peak normalized firing rate with Oxo-M versus the basal firing rate of control and NALCN KO SNr neurons. Red lines are linear regression fits of the data. Control: $p$=0.2; NALCN KO: $p$=0.6; Pearson correlation. (**E**) Baseline and peak firing rate in Oxo-M of WT, TRPC3 KO and hepta-TRPC KO SNr neurons. (**F**) Peak fold increase in firing rate with Oxo-M of WT (2.7 ± 0.3; $n$ = 5), TRPC3 KO (4.5 ± 1.0; $n$ = 7) and hepta-TRPC KO neurons (4.1 ± 1.2; $n$ = 3). [n.s.]p>0.05; one-way ANOVA.

## Other highly expressed NSCCs do not contribute to the spontaneous activity of SNr neurons

While conditional knockout of NALCN significantly diminished the spontaneous firing of SNr neurons, SNr neurons were not completely silenced, indicating that other nonselective cation channels may contribute to maintaining spontaneous firing. Our RNA-seq results identified additional nonselective cation channel candidates, including channels that have not been previously investigated in the SNr. Although TRPML1 was the next highest expressed NSCCs candidate in our sequencing results (*Figure 3B*), TRPML channels are not found in the plasma membrane and likely do not contribute to the membrane potential in neurons (*Cheng et al., 2010*). We also did not focus on HCN channels as it has been previously shown that inhibitors of HCN channels are without effect on SNr firing (*Atherton and Bevan, 2005*).

We proceeded to test whether perturbing the function of other highly expressed NSCCs for which blockers or genetic tools were available could modify SNr firing. We first tested whether TRPM7, which is known to function as a channel in the plasma membrane (*Bates-Withers et al., 2011*), was involved in maintaining spontaneous firing by conditionally ablating TRPM7 in SNr neurons. We used a transgenic *Trpm7[fl/fl]* mouse line (*Jin et al., 2008*) to virally transduce neurons with the hSyn-EGFP-cre vector. We found that knockout of TRPM7 was without effect on SNr firing (*Figure 8A*), arguing against a role of TRMP7 in the spontaneous activity of SNr neurons.

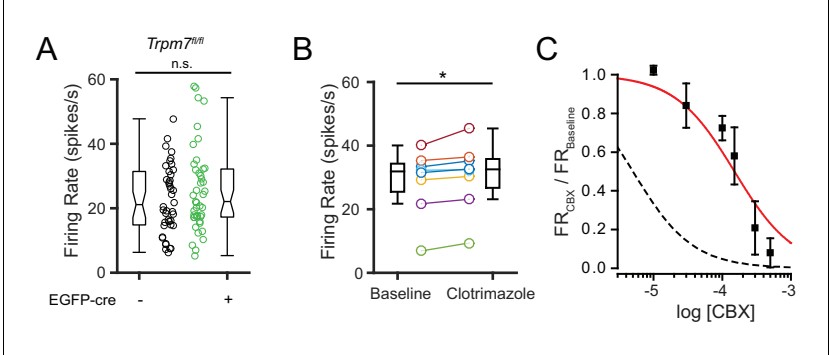

**Figure 8.** Pharmacological and genetic tests of other candidate NSCC. (**A**) Mean firing rate of control (23.5 ± 1.9; $n$ = 42) and TRPM7 KO (25.8 ± 2.0; $n$ = 43) SNr neurons. [n.s.]$p$=0.42; unpaired t-test. (**B**) Mean firing rate before (28.8 ± 3.6) and after (30.6 ± 3.7) application of a TRPM2 channel blocker, clotrimazole (10 μM). *$p$=0.01; paired t-test. (**C**) Dose response curve of the effect of carbenoxolone (CBX) on baseline normalized firing rate (red line; Hill function) compared with predicted dose response curve of CBX blockade of pannexin channels (dashed black line; simulated Hill function; k = 5 × $10^{-6}$M).

We next tested the involvement of TRPM2, another NSCC we found to be expressed by SNr neurons. Previous studies have shown that TRPM2 is activated by NMDA receptors or by hydrogen peroxide leading to elevated firing of SNr neurons (*Lee et al., 2013*). However, no evidence of constitutively open TRPM2 channels has been reported in the SNr, making it unlikely that this channel supports the spontaneous firing. We found that rather than decreasing SNr firing rate, a blocker of TRPM2, clotrimazole (*Hill et al., 2004*), produced a significant but very small increase in firing rate (*Figure 8B*). These results do not support a substantial role for TRPM2 in baseline firing.

Lastly, we tested whether pannexin channels (PANX1 and PANX2) are involved in the basal firing rate of SNr neurons. These channels are inhibited by carbenoxolone (CBX), but this drug can block other ion channels as well (*Connors, 2012*). Inhibition by CBX at relatively lower concentrations can be indicative of a role of pannexin channels (*Ma et al., 2009*; *Bruzzone et al., 2005*). Therefore, we tested several concentrations of CBX on SNr spontaneous firing and found a dose-dependent inhibition of firing (*Figure 8C*). However, the observed $IC_{50}$ was much higher ($K_d$ = 1.5 × $10^{-4}$ M) than expected for a pannexin-dependent effect ($K_d$ = 5 × $10^{-6}$ M), arguing against an important role of pannexin channels. Overall, these results suggest that NALCN is the main NSCC that contributes to spontaneous firing of SNr neurons, while the other highly expressed NSCCs are not critically involved in spontaneous firing.

## Discussion

Many neuron types display pronounced, intrinsically-regulated spontaneous firing patterns that are crucial for normal circuit function (*Häusser et al., 2004*; *Bean, 2007*). However, the molecular machinery that sustains this firing is not fully understood (*Häusser et al., 2004*; *Ren, 2011*). "Leak current" has long been a catch-all term to describe the depolarizing current that is one component of spontaneous firing (*Atherton and Bevan, 2005*; *Raman et al., 2000*; *Jackson et al., 2004*; *Taddese and Bean, 2002*; *Khaliq and Bean, 2010*). Spontaneously firing GABAergic SNr neurons provide a key source of inhibitory tone to target areas including the superior colliculus and pedunculopontine nucleus (*Hikosaka, 2007*). In SNr neurons, a sodium-dependent, TTX-insensitive depolarizing current has been attributed to a NSCC, but the molecular identity of this current remains undefined (*Atherton and Bevan, 2005*). Identifying the molecular basis of this current is important to enable detailed investigation of how its properties and regulation underlie SNr tonic inhibition of basal ganglia targets.

In order to address this, we performed whole transcriptome sequencing of SNr neurons and detected expression of several NSCCs that could potentially generate the tonic depolarizing current. We discovered that SNr neurons express NALCN, which stood out as a prime candidate for the

channel carrying the depolarizing leak current in these cells. Using a combination of pharmacological and genetic techniques, we showed that: 1) NALCN is functionally active in SNr neurons; 2) NALCN is an important component of the machinery that maintains spontaneous firing in SNr neurons, 3) NALCN is required for sensitivity of SNr neuron firing to glycolysis, and 4) NALCN is important for muscarinic receptor-dependent stimulation of SNr neuron firing. Taken together, our findings support a critical role of NALCN in both maintenance of spontaneous firing of SNr neurons and physiological modulation of SNr neuron excitability.

NALCN is highly conserved evolutionarily. In the invertebrate species *D. melanogaster*, *C. elegans*, and *L. stagnalis*, their respective NALCN homologs are critical for pacemaker neuron function in circadian rhythms (*Nash et al., 2002*; *Lear et al., 2005*), locomotion (*Yeh et al., 2008*; *Gao et al., 2015*), and respiration (*Lu and Feng, 2011*). In mice, germ-line ablation of NALCN produces animals that develop normally, but die soon after birth from impairment in the rhythmically active neurons that regulate respiration (*Lu et al., 2007*). Hippocampal neurons cultured from NALCN KO mice show significantly hyperpolarized membrane potentials indicating an important function of NALCN in maintaining depolarized membrane potentials. A recent study employing conditional knockout of NALCN in forebrain excitatory neurons demonstrated that NALCN is necessary for appropriate excitability of neurons of the suprachiasmatic nucleus and underlies circadian rhythms in flies and rodents (*Flourakis et al., 2015*). In addition, this study reported that the forebrain NALCN knockout mice died $\sim$21 days after birth, further indicating the critical function of this channel. Our results demonstrate that NALCN is necessary for normal spontaneous firing of SNr neurons and for the regulation of SNr excitability. The combined results of these studies are consistent with the idea that NALCN is a critical component of neuronal excitability in many rhythmically active neurons (*Ren, 2011*).

NALCN also appears to mediate several modulatory influences on SNr firing. We find that it is important for regulation of SNr neuronal excitability in response to changes in glycolysis, an effect we had seen previously when investigating metabolic influences on SNr firing rate (*Lutas et al., 2014*). Together with our finding that the increased firing produced by muscarinic receptor activation is blunted in NALCN KO SNr neurons, these results suggest that tuning the level of NALCN activity may set both the tonic and phasic excitability of SNr neurons.

Interestingly, NALCN has also been found in insulin-releasing pancreatic beta cells, and muscarinic receptor activation can increase NALCN activity in these cells (*Swayne et al., 2009*). Our finding that NALCN activity decreases when we block glycolysis in SNr neuron may also occur in beta cells. In beta cells, a decrease in NALCN activity along with an increase in ATP-sensitive potassium channel activity when glucose levels are low would work together to hyperpolarize beta cells and prevent insulin release. Therefore, the metabolic sensitivity of NALCN may be a broadly used modality to confer metabolic sensitivity to NALCN-expressing cells.

One question that arises from our results is how the activity of NALCN is decreased by the inhibition of glycolysis. A possible insight comes from the finding that metabotropic receptor signaling can modulate the activity of NALCN in neurons (*Lu et al., 2009*) and pancreatic beta cells (*Swayne et al., 2009*). Interestingly, this metabotropic receptor signaling does not modulate NALCN via canonical G-protein coupled signaling; rather, channel activity is regulated by phosphorylation of the channel by Src kinases (*Ren, 2011*). Phosphorylation of NALCN may be the major regulator of NALCN activity, either activating or inhibiting channel activity. During inhibition of glycolysis, the drop in cytosolic [ATP] and increase in [AMP] activates AMP kinase (AMPK) (*Kahn et al., 2005*), which may directly phosphorylate NALCN at a site that leads to decreased NALCN activity. An example of this possibility is the ion channel Cystic Fibrosis Transmembrane Conductance Regulator (CFTR), which is activated by Src kinases phosphorylation (*Billet et al., 2015*), but inhibited by AMPK phosphorylation (*Hallows et al., 2000*; *Kongsuphol et al., 2009*). Alternatively, there may be crosstalk between AMPK and the ability of Src family kinases to activate NALCN (such crosstalk is seen in regulation of metabolism by the Src family kinase Fyn and AMPK (*Bastie et al., 2007*)). An exciting direction for future studies will be to better understand how intracellular signaling pathways regulate NALCN activity in neurons.

Our findings may also have clinical relevance, as mutations in NALCN have been implicated in several human disorders (*Cochet-Bissuel et al., 2014*). Recent studies have found that mutations in NALCN result in severe motor and cognitive deficits in humans (*Chong et al., 2015*; *Aoyagi et al., 2015*; *Al-Sayed et al., 2013*; *Köroğlu et al., 2013*; *Fukai et al., 2016*). In addition, humans carrying

mutations in UNC80, a protein that is important for the function of NALCN (*Ren, 2011*), show severe deficits that mirror those observed in humans with mutations in NALCN (*Perez et al., 2015*; *Shamseldin et al., 2016*; *Stray-Pedersen et al., 2016*). The SNr is a main output of the basal ganglia motor circuit and provides tonic inhibition of target areas (*Hikosaka, 2007*). Our finding that SNr neurons require NALCN for appropriate tonic firing, suggests that NALCN mutations could contribute to basal ganglia related motor disorders. For example, disruption of the normal tonic firing of SNr neurons by injection of GABA receptor agonists, leads to excessive muscle tension and dystonia (*Burbaud et al., 1998*). In addition, hyperactivity of the SNr is observed in Parkinsonian animals (*Wichmann et al., 1999*) and ablation of the SNr can ameliorate symptoms of Parkinson's diseases (*Wichmann et al., 2001*). Therefore, understanding the function of NALCN in modulating the tonic activity of SNr neurons may lead to a more detailed mechanistic understanding of motor dysfunction in humans and potentially provide a new therapeutic target for motor disorders.

While NALCN is a major contributor to the spontaneous firing rate of SNr neurons, elimination of NALCN resulted in only a halving in the firing rates of SNr neurons. This observation is in contrast to the effects of $Gd^{3+}$, which completely silenced firing of WT SNr neurons; this suggests that other $Gd^{3+}$-sensitive currents, in addition to NALCN, may drive the firing of SNr neurons. We tested the possibility that other NSCCs that we found to be expressed in SNr neurons (TRPM7, TRPM2, and pannexin) may contribute to the firing rate; however, we found that TRPM7 knock-out or pharmacological inhibition of TRPM2 or pannexin were without effect on the firing rate of SNr neurons. Therefore, we were unable to identify NSCCs, other than NALCN, that are critical for spontaneous firing. One interesting possibility to explain the continued firing in the NALCN KO is that the expression of inward-current-carrying leak channels may be balanced homeostatically with outward-current-carrying leak potassium channels. In that case, loss of NALCN would lead to decreased expression of leak potassium channels and therefore offset the decreased inward current, allowing spontaneous firing to persist. While this idea is attractive, we did not find direct evidence of this possibility. In our single-cell gene expression dataset, there was no significant correlation between the expression of NALCN and the two highest expressed leak potassium channels, KCNK1 and KCNK3. Additionally, the results of our steady-state I-V recordings were inconsistent with a decrease in leak potassium current. If outward leak currents were decreasing along with inward leak currents, we would expect to observe a significant increase in the membrane resistance of SNr neurons lacking NALCN, but we did not. It remains possible that changes in the expression of other ion channels may compensate for the loss of NALCN in the knockout, as spontaneous firing is a critical function of the SNr.

## Materials and methods

### Animals

The Harvard Medical Area Standing Committee on Animals approved all procedures involving animals. Brain slice electrophysiology experiments were performed using brains of male and female 2 to 3 week old wild-type (WT) mice (C57/BL6; Charles River Laboratories), *Nalcn*<sup>fl/fl</sup> mice (*Flourakis et al., 2015*), and *Trpm7*<sup>fl/fl</sup> mice (*Jin et al., 2008*). TRPC3 knockout mice and mice lacking all seven TRPC channels (TRPC1-7) were previously described (*Lutas et al., 2014*). GAD1-EGFP mice (*Tamamaki et al., 2003*) were used for single-cell SNr neuron collection.

### Acute brain slice preparation

Preparation of brain slices was performed as previously described (*Lutas et al., 2014*). Briefly, mice were first anesthetized via isoflurane inhalation and then decapitated. Using a vibrating tissue slicer (Campden, Lafayette, IN; 7000smz-2), we made acute coronal midbrain slices (275 µm) containing the substantia nigra region. Coronal slices (typically 2–3 slices per animal) were hemi-sectioned to obtain 6 total slices containing the SNr region. All slicing procedures were performed in ice-cold slicing solution followed by immediate incubation in ACSF at 37°C for 35 min. Slices were then kept at room temperature in ACSF for 25 min to 3 hr before being used for recording. Slicing solution and ACSF were continuously oxygenated with 95% $O_2$ and 5% $CO_2$.

## Solutions

The solutions used in this study were similar to those previously described (*Lutas et al., 2014*). Slicing solution consisting of (mM): 215 sucrose, 2.5 KCl, 24 $NaHCO_3$, 1.25 $NaH_2PO_4$, 0.5 $CaCl_2$, 7 $MgCl_2$, 10 D-glucose ($\sim$310 mOsm, pH = 7.4). ACSF consisted of (mM): 125 NaCl, 2.5 or 4 KCl, 25 $NaHCO_3$, 1.25 $NaH_2PO_4$, 1.5 $CaCl_2$, 1 $MgCl_2$, 10 D-glucose ($\sim$300 mOsm, pH = 7.4). For experiments using the mitochondrial fuel βHB (2.5 mM), we used sodium (R)-3-hydroxybutyrate, which is the specific enantiomer of βHB that can be metabolized. The pipette solution for loose patch cell-attached recordings consisted of (mM): 150 NaCl, 2.5 KCl, 10 HEPES, 1.5 $CaCl_2$, 1 $MgCl_2$ ($\sim$300 mOsm; pH 7.4). For whole-cell recordings, the pipette solution consisted of the following (in mM): 140 K-gluconate, 10 NaCl, 10 HEPES, 1 $MgCl_2$, and 0.1 EGTA, pH 7.3 ($\sim$300 mOsm)

For the collection of single-cells, dissociated tissue was maintained in a HEPES-based ACSF (mM): 200 sucrose or 150 NaCl, 6 $MgSO_4$, 2 KCl, 0.5 $CaCl_2$, 10 HEPES, 10 Glucose. When NaCl was used, 1 mM Na-lidocaine was included. pH was adjusted to 7.4 with NaOH. 1% fetal bovine serum (FBS) was added to the solution shortly before the experiment. All solutions used for dissociation and collections of cells were filtered using 0.2 μm syringe filters (VWR, Philadelphia, PA).

## Electrophysiology

The spontaneous firing rate of GABAergic neurons of the SNr was monitored using loose-patch cell-attached recordings as previously described (*Lutas et al., 2014*). Briefly, we used borosilicate pipettes (Warner Instruments, Hamden, CT) which had tip resistances of $\sim$2 MΩ when filled with HEPES-buffered ACSF solution. Whole-cell voltage-clamp recordings to characterize current–voltage (I–V) relationships were performed as previously described (*Lutas et al., 2014*). Voltage steps (150 ms duration) were made from an initial holding potential of -30 mV in 10 mV decrements, and steady-state current was calculated from the average of a 10 ms window at the end of each 150 ms voltage step. Experiments were completed within 1 min of breaking into the neuron. Reported voltages have been corrected for liquid junction potentials of 15 mV. Pipettes had tip resistances of 2–4 MΩ.

All recordings were performed at 34°C with continuous perfusion (flow rate: 5 mL/min) in a custom made dual-perfusion chamber. Neurons were visualized using an upright microscope (BX51WI; Olympus, Center Valley, PA) equipped with IR-DIC and controlled using TILL Vision (TILL Photonics, Grafelfing, Germany). All recordings were collected and amplified via a Multiclamp 700B (Molecular Devices, Sunnyvale, CA). Recordings were low-pass filtered at 4 kHz and sampled at 10 kHz. Signals were digitized using a Digidata 1321A (Molecular Devices) and acquired using pClamp 10 (Molecular Devices).

## Pharmacology

All chemicals used were obtained from Sigma-Aldrich (St. Louis, MO) or Tocris Bioscience (Bristol, UK). Recordings were performed in the continuous presence of synaptic blockers of ionotropic glutamate (kynurenic acid; 1 mM) and GABA receptors (picrotoxin; 100 μM) to eliminate spontaneous synaptic events. No difference was observed in basal firing rates between experiments performed in the presence or absence of synaptic blockers, consistent with previous findings (*Atherton and Bevan, 2005*). All hydrophobic drugs were dissolved in DMSO to obtain stock solutions. Final DMSO concentrations in ACSF were <0.1% and this concentration of DMSO had no effect on SNr spontaneous firing. Inhibition of glycolysis was achieved using iodoacetic acid (IAA) as previously described (*Lutas et al., 2014*). For whole cell recordings, sodium lidocaine (1 mM) and tetraethylammonium chloride (1 mM) were included in the bath solution.

## Data analysis

All analysis was performed using Matlab (R2014b; Mathworks, Natick, MA). Electrophysiological recordings were digitally high-pass filtered at 1 Hz and a threshold was used to detect individual action potentials in bins of one second. For drug applications, 60 s in each condition (baseline or drug) were used as a measure of mean firing frequency. For normalized firing rate plots, individual experiments were normalized to a 60 s baseline average. Graphs depicting the time course of firing rate with drug application were smoothed using a 10 s moving window average. For whole cell I-V recordings, all recordings analyzed had input resistances greater than 120 MΩ and access

resistances less than 25 MΩ. Descriptive statistics are reported as mean ± SEM. Sample size reported indicates number of neurons. For comparisons between two populations, paired or unpaired two-tail Student's *t*-test was used. For multiple comparisons, one-way ANOVA with Bonferroni *post hoc* test with alpha = 0.05 was used.

For dose response curves, a Hill function was fit to the data using Origin 9.1 (Origin Lab, Wellesley, MA). The maximum and minimum values for the Hill function were fixed to 1 and 0, and the Hill coefficient was fixed to 1.

## Manual single-cell collection for RNA-seq

Collection of single SNr neurons was performed using a protocol modified from a previously published manual cell sorting method (*Hempel et al., 2007*). Coronal mouse brain slices (275 μm) were prepared from GAD1-EGFP mice. Slices containing the substantia nigra region were incubated for 30 min at room temperature in slicing solution bubbled with 95% $O_2$/ 5% $CO_2$, which also contained a proteolytic enzyme (Pronase E, 2 mg/mL, P5147 Sigma). To silence neuronal firing and prevent excitotoxicity, slices were then rinsed with HEPES-based ACSF containing 1% FBS and which either had NaCl completed replaced by equiosmolar sucrose or, if NaCl was not removed, the ACSF contained the sodium channel blocker Na-lidocaine (1 mM).

The region containing the SNr was microdissected under a dissecting microscope from two brain slices, which produced four total SNr pieces per mouse. These SNr regions were placed into an Eppendorf tube containing 500 μL of HEPES-based ACSF with 1% FBS. The tissue was then gently triturated with fire-polished Pasteur pipettes (230 mm; Wheaton, Millville, NJ) of progressively smaller tip diameter (400, 250, 150 μm). Typically less than 10 passages were performed with each Pasteur pipette and these were performed slowly to avoid formation of bubbles. The solution contained the dissociated tissue was then diluted in 20 mL of HEPES-based ACSF with 1% FBS and poured into a sterile 140 mm petri dish (VWR). The petri dish was placed in an inverted fluorescence microscope (Nikon, Tokyo, Japan; Eclipse TE300) to detect GFP positive cells. Individual cells were collected by aspiration into unfilamented borosilicate pipettes (Warner Instruments, G150-4) pulled to have tip diameters around the size of a cell soma (10–30 μm). Pipettes were positioned using a micromanipulator (Sutter Instruments, Novato, CA; MPC100) while visually monitoring with 160x magnification. Pipettes were not pre-filled with solution and slight positive pressure was maintained when entering the solution in the petri dish to prevent solution from entering the pipette. Unfilamented capillary tubes were used to minimize capillary action. Typically only a few millimeters of solution entered the pipette tip and the meniscus of the solution was clearly visible in the field of view.

When a GFP positive cell was identified, the pipette was rapidly lowered into the solution and brought into close apposition to the cell. Slight suction via syringe was applied to gently capture the cell and the pipette was immediately withdrawn from the solution. The pipette tip was broken into the bottom of a PCR tube contained 5 μL of TCL-buffer (Qiagen) and pressure was applied via syringe to eject any solution in the pipette. The PCR tube was then flash frozen in ethanol with dry ice and stored at −80°C until being sent for library preparation and sequencing. Around 15 cells were collected from each dissociated cell preparation and this collection was completed in 1–2 hr. A total of 96 cells were collected for sequencing.

## Single-cell RNA sequencing

Individual neurons were lysed in 5 μL of TCL buffer (Qiagen), and cDNA libraries were prepared by the Broad Technology Labs (BTL) and sequenced by the Broad Genomics Platform. Libraries were prepared according to the Smart-Seq2 protocol (*Picelli et al., 2013*; *2014*) with some modifications (*Trombetta et al., 2014*). Briefly, total RNA was purified using RNA-SPRI beads and poly(A)-tail mRNA was converted to cDNA before being amplified. cDNA was subject to transposon-based fragmentation that used dual-indexing to barcode each fragment of each converted transcript with a combination of barcodes specific to each sample. Barcoded cDNA fragments were then pooled prior to sequencing. Paired-end reads (2 x 25 bp) were generated using an Illumina Next-Seq500 (San Diego, CA).

## Analysis of sequencing results

Data was initially processed through the BTL pipeline: data was separated by barcode and aligned using Tophat version 2.0.10 (*Kim et al., 2013*) to the mouse genome (Genome Reference Consortium GRCm38). Transcripts were quantified using the Cufflinks suite version 2.2.1 (*Trapnell et al., 2012*). Data was processed if 50% of the reads align and there were at least 100,000 aligned pairs per cell. Data was initially normalized as fragments per kilobase pair per million mapped reads (FPKM) and then converted to the number of transcripts per million (TPM). The gene expression results are reported as the log2 transform of the TPM + 1. A gene was considered expressed by an individual sample if TPM > 1. Only samples with greater than 5000 genes detected were used for analysis. For determining the consistently expressed ion channel genes, only ion channels detected in at least 50% of samples were included in the cumulative distribution plot.

## Injections of virus for conditional knockout

Stereotaxic guided surgeries were performed in mice postnatal day 7 or 8. Mice were anesthetized with isoflurane (induction: 4–4.5%; maintenance: 2–3% ). Local anesthetics (50 µL of a 1:1 mixture of 0.25% lidocaine and 0.0625% bupivacaine) were injected at the incision site. The analgesic ketoprofen (10 mg/kg) was injected subcutaneously at the beginning of the surgery. Body temperature was maintained at 37°C by a heated pad below the mouse. The skull was exposed via a small incision and a small hole was drilled to target the SNr. SNr coordinates (from lambda in mm): 1.1 anterior, -1.1 lateral, -3.6 from pia. To conditionally disrupt *Nalcn* or *Trpm7*, we injected mice with a virus carrying an EGFP tagged cre construct (AAV9.hSyn.HI.eGFP-Cre.WPRE.SV40; catalog number AV-9-PV1848; UPenn Viral Vector Core). The virus was diluted in saline to a final titer of $5.54 \times 10^{11}$ genome copies/mL. Using a pulled glass capillary pipette (Wiretrol II, Drummond Scientific Company, Broomall, PA) connected to an UltraMicroPump III (WPI, Sarasota, FL) microinjector, 200 nL of virus was injected at a rate of 100 nL/min. The pipette was left in place for 5 min before removal. We waited 1–2 weeks post-injection before performing brain slice electrophysiology from these mice.

## Acknowledgements

We thank members of the Yellen lab for assistance and comments. We are also grateful to Drs. Bruce Bean, Michael Do, Chinfei Chen, Bernardo Sabatini, David Clapham and Mark Andermann for advice and comments. We thank Drs. Dejian Ren, Ravi Allada and Matthieu Flourakis for sharing the floxed *Nalcn* transgenic mice, and Drs. Shu-Hsien Sheu, Sunday Abiria and David Clapham for sharing the floxed *Trpm7* transgenic mice. The TRPC3 and sevenfold TRPC knock-out mice were generously provided by Dr. Lutz Birnbaumer. We thank Dr. Bernardo Sabatini for providing the GAD1-EGFP mice. We thank the Harvard Neurobiology Imaging Facility (NINDS P30 Core Center grant #NS072030) for instrument availability that supported this work. This work was supported by NIH/NINDS grants R01 NS055031 to GY and F31 NS077633 to AL, and by NIH Office of the Director grant DP1 EB016985 to GY.

## Additional information

### Funding

| Funder | Grant reference number | Author |
| --- | --- | --- |
| National Institute of Neurological Disorders and Stroke | R01 NS055031 | Andrew Lutas<br>Gary Yellen |
| National Institutes of Health | DP1 EB016985 | Carolina Lahmann<br>Gary Yellen |
| National Institute of Neurological Disorders and Stroke | F31 NS077633 | Andrew Lutas |

The funders had no role in study design, data collection and interpretation, or the decision to submit the work for publication.

### Author contributions
AL, Conception and design, Acquisition of data, Analysis and interpretation of data, Drafting or revising the article; CL, MS, Acquisition of data, Drafting or revising the article; GY, Conception and design, Analysis and interpretation of data, Drafting or revising the article

### Author ORCIDs
Andrew Lutas, http://orcid.org/0000-0002-6991-2898
Gary Yellen, http://orcid.org/0000-0003-4228-7866

### Ethics
Animal experimentation: This study was performed in strict accordance with the recommendations in the Guide for the Care and Use of Laboratory Animals of the National Institutes of Health. All experimental manipulations were performed in accordance with protocols approved by the Harvard Medical Area Standing Committee on Animal Care (#03506).

# Additional files

## Major datasets

The following dataset was generated:

| Author(s) | Year | Dataset title | Dataset URL | Database, license, and accessibility information |
|---|---|---|---|---|
| Lutas A, Lahmann C, Soumillon M, Yellen G | 2016 | Transcriptomes of individual substantia nigra pars reticulata neurons | http://www.ncbi.nlm.nih.gov/geo/query/acc.cgi?acc=GSE78521 | Publicly available at the NCBI Gene Expression Omnibus (accession no: GSE87521). |

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
