## [Decision Letter]

Thank you for submitting your article "The leak channel NaLCN controls tonic firing and glycolytic sensitivity of substantia nigra pars reticulata neurons" for consideration by *eLife*. Your article has been reviewed by three peer reviewers, one of whom is a member of our Board of Reviewing Editor, and the evaluation has been overseen by Richard Aldrich as the Senior Editor.

The reviewers have discussed the reviews with one another and the Reviewing Editor has drafted this decision to help you prepare a revised submission.

The following individuals involved in review of your submission have agreed to reveal their identity: Dejian Ren (peer reviewer).

Summary:

This manuscript identifies NALCN as the primary non-selective cation channel that drives spontaneous firing in GABAergic substantia nigra pars reticulata (SNr) neurons. The spontaneous firing or "pacemaking" properties of neurons are of interest, because knowing which ionic currents drive repetitive firing will not only help explain how these neurons code information but might also identify possible therapeutic targets, since disruptions of basal activity are associated with disease states. Here, the authors rule out some candidate ion channels for making SNr neurons fire, and then take the more comprehensive approach of doing RNA-seq and finding high expression of NALCN, the sodium leak conductance channel. This work shows that Nalcn mRNA is highly expressed in SNr neurons and also demonstrates the glycolysis sensitivity of channel function as well as the susceptibility to modulation by neurotransmitters. It thus reveals NALCN as a fundamental yet tunable regulator of excitability in central neurons.

Essential revisions:

The reviewers expressed uniform enthusiasm for the work; these are included as "general comments". (6) The primary issue raised by the reviewers was the value of having some kind of direct measure of the NALCN current, e.g., under voltage clamp, to demonstrate the modulation and/or to confirm changes with viral knockdown. (18) In addition, a few points were raised that would be good to address in the Discussion: (a) hints from the present data set about the role of other inward currents (from Gd sensitivity) or lack of outward currents (from expression data) in facilitating firing and (b) mechanisms of modulation by glycolysis.

General comments from the three reviewers.

Reviewer 1: This is a high-quality and clearly written manuscript identifying NaLCN as the primary non-selective cation channel that drives spontaneous firing in GABAergic substantia nigra neurons. The spontaneous firing or "pacemaking" properties of various neurons are of interest, especially those in the basal ganglia (here, the substantia nigra pars reticulata SNr) because disruptions of basal activity are associated with diseases like Parkinson's. Knowing what ionic currents drive repetitive firing will not only help explain how these cells code information but might also identify possible therapeutic targets. Here, the authors rule out some obvious candidates for what makes SNr neurons fire, and then take the more comprehensive approach of doing RNA-seq and finding high expression of the fairly recently discovered NaLCN, the sodium leak conductance channel. The data are convincing and well presented. This manuscript goes beyond previous physiological work from other labs (on other neurons), not only in showing at the RNA level that NACLN is highly expressed in SNr neurons but also in demonstrating the glycolysis sensitivity of channel function, and the susceptibility to modulation by neurotransmitters. It thus sets up NALCN as a fundamental yet tunable regulator of excitability in central neurons.

Reviewer 2: This paper addresses a very important neurobiology question: why some neurons spontaneously fire? It has been known for decades that, in many spontaneously firing neurons, a tonically active cation conductance "constantly" drives the membrane potential toward threshold to generate spontaneous firing. A commonly held model, although rarely tested, is that some of the non-selective TRP channels code the cation conductance. With several elegant approaches, the authors in this paper overthrow this model and establish a new paradigm. To "unbiasedly" identify the conductance, the authors first painstakingly obtained single cell transcriptomes of 62 identified SNr neurons (known to spontaneously fire). With this data set (which by itself is a useful resource to the community), the authors fished out all the ion channel genes with high expression in the neurons. Guided by the finding that NALCN and its associated subunits UNC79 and UNC80 are the highest expressors, the authors used pharmacological tools and a Nalcn conditional knockout to demonstrate that NALCN indeed is a key to spontaneous firing in the SNr neurons. In addition, they discovered that NALCN is required for the normal regulation of firing rates by glycolysis inhibition and by mAChR activation. Finally, the authors provide data against the TRP-based model: inhibiting or knocking out several TRP channels had no major effect on the spontaneous firing. The paper establishes the long-sought molecular identity of a conductance essential for a highly important behavior of some neurons. It also clears up a common misconception in the field. The experiments are well done and the conclusions are convincing.

Reviewer 3: This elegant paper is a continuation of ongoing research on the mechanism of spontaneous firing of the GABAergic neurons in the substantia nigra pars reticulata (SNr). First, authors developed a method to identify, isolate, and collect samples from SNr GABAergic neurons. Then, using transcriptome analysis, the determined all expressed nonselective cation channels (NSCCs) channels according to their weight in the whole transcriptome expression. Then, they functionally evaluate channels with the highest expression. One of the strongest points is that authors clearly demonstrate that controversial TRPCs channels, which previously thought to modulate SNr neuronal firing, are not involved in it. The data clearly point out sodium NaLCN leak channel as a primary mediator of the SNr neuronal tonic firing, based on intersectional pharmacology and knock down experiments. Authors evaluated a number of other highly expressed channels, but found none of them to be involved in the neuronal firing modulation. Furthermore they document regulation of NaLCN channels by glycolytic pathways and mGluRs. The paper is well written, reasonably organized and easy to read. It has outstanding scientific value and substantially contributes to the field of neuroscience, specifically to our understanding of mechanisms of spontaneous neuronal activity. The NaLCN – dependent mechanism demonstrated in this work could lead to a development of new drugs targeting NaLCN channel.

---

## [Author Response]

The reviewers expressed uniform enthusiasm for the work; these are included as "general comments". (6) The primary issue raised by the reviewers was the value of having some kind of direct measure of the NALCN current, e.g., under voltage clamp, to demonstrate the modulation and/or to confirm changes with viral knockdown. (18) In addition, a few points were raised that would be good to address in the Discussion: (a) hints from the present data set about the role of other inward currents (from Gd sensitivity) or lack of outward currents (from expression data) in facilitating firing and (b) mechanisms of modulation by glycolysis.

We thank the reviewers for their careful reading of our manuscript and appreciate the enthusiasm for our work. To address the primary issue raised by the reviewers, we have performed whole-cell voltage-clamp recordings in NALCN KO SNr neurons, and these results provide additional support for the role of NALCN channels in providing tonic drive for the SNr neurons. We have also addressed the points raised about additional discussion topics.